# Warsaw Glacial Quartz Sand with Different Grain-Size Characteristics and Its Shear Wave Velocity from Various Interpretation Methods of BET

**DOI:** 10.3390/ma14030544

**Published:** 2021-01-23

**Authors:** Katarzyna Gabryś, Emil Soból, Wojciech Sas, Raimondas Šadzevičius, Rytis Skominas

**Affiliations:** 1Water Centre WULS, Warsaw University of Life Sciences, 02-787 Warsaw, Poland; wojciech_sas@sggw.edu.pl; 2Institute of Civil Engineering, Department of Geotechnics, Warsaw University of Life Sciences, 02-787 Warsaw, Poland; emil_sobol@sggw.edu.pl; 3Institute of Hydraulic Engineering, Vytautas Magnus University Agriculture Academy, 53361 Kaunas, Lithuania; raimondas.sadzevicius@vdu.lt (R.Š.); rytis.skominas@vdu.lt (R.S.)

**Keywords:** shear wave velocity, sand, bender elements test, grain-size characteristics

## Abstract

After obtaining the value of shear wave velocity (*V_S_*) from the bender elements test (BET), the shear modulus of soils at small strains (*G_max_*) can be estimated. Shear wave velocity is an important parameter in the design of geo-structures subjected to static and dynamic loading. While bender elements are increasingly used in both academic and commercial laboratory test systems, there remains a lack of agreement when interpreting the shear wave travel time from these tests. Based on the test data of 12 Warsaw glacial quartz samples of sand, primarily two different approaches were examined for determining *V_S_*. They are both related to the observation of the source and received *BE* signal, namely, the first time of arrival and the peak-to-peak method. These methods were performed through visual analysis of BET data by the authors, so that subjective travel time estimates were produced. Subsequently, automated analysis methods from the GDS Bender Element Analysis Tool (BEAT) were applied. Here, three techniques in the time-domain (TD) were selected, namely, the peak-to-peak, the zero-crossing, and the cross-correlation function. Additionally, a cross-power spectrum calculation of the signals was completed, viewed as a frequency-domain (FD) method. Final comparisons between subjective observational analyses and automated interpretations of BET results showed good agreement. There is compatibility especially between the two methods: the first time of arrival and the cross-correlation, which the authors considered the best interpreting techniques for their soils. Moreover, the laboratory tests were performed on compact, medium, and well-grained sand samples with different curvature coefficient and mean grain size. Investigation of the influence of the grain-size characteristics of quartz sand on shear wave velocity demonstrated that *V_S_* is larger for higher values of the uniformity coefficient, while it is rather independent of the curvature coefficient and the mean grain size.

## 1. Introduction

Shear wave (S-wave) velocity (*V_S_*) is a very important parameter in the field of geotechnical earthquake engineering [1,2]. The magnitude of shear wave velocity is determined from both in situ and laboratory tests, and this is used for computing the low-strain shear modulus (*G_max_*) [3]. *G_max_* is an essential input parameter for dynamic stability analysis of slopes, embankments, dams, etc. The most versatile and portable method to assess *G_max_* in the laboratory is the study of shear wave propagation through bender elements (BE) tests or the ultrasonic method [4]. The other group of methods is based on vibrations such as torsional shear (TS) and resonant column (RC) tests [5]. 

This paper is dedicated to the bender elements test (BET) as a non-destructive test that has gained popularity in the laboratory determination of the low-strain shear modulus, due to its simplicity in use, low cost, and aforementioned non-destructive operation [6]. Bender elements are made of two piezoelectric ceramic sheets with a central shim of usually ferrous nickel alloys to enhance its strength. Bender elements operate as an electromechanical transducer. When a small voltage is applied to one element, it will bend due to the polarization that has induced across its plates, and thus it will act as a transmitter element. On the contrary, when an element bends, a voltage is generated and so it will act as a receiver one [7]. As the strain induced in soil due to bender element movement is in the elastic strain range of soils behavior (<10^−3^%), based on the theory of elastic wave propagation [8], the low-strain shear modulus can be derived from Equation (1):(1)$$G_{max}=\rho \cdot V_{S}^{2}$$
where *ρ* is soil density and *V_S_* is shear wave velocity. The transmitter and receiver elements are inserted at the two opposite ends of soil specimens and the time lag(*t*), between the input and output wave signals, is measured by triggering and propagating a wave impulse through the soil specimen at a certain frequency and voltage [9]. Shear wave velocity is then computed using Equation (2):(2)$$V_{S}=\frac{L_{tt}}{t}$$
where *L_tt_* is the tip-to-tip distance between transmitter and receiver [10]. 

Research methods with the use of piezoelements, although known and applied for several decades [11], have not received any formal standards. They were developed according to the criteria created for the needs of subsequent research centers and may not necessarily be universally adopted. The multitude of design solutions for apparatus does not help in unification. In addition to the few solutions available commercially, there are many devices constructed mainly in research centers. Apart from the same main principle of operation, these apparatuses differ quite significantly in terms of detailed hardware solutions and software. To systematize this issue and try to estimate the impact of the methodology on results under the auspices of the Japanese Technical Committee TC-29, a parallel study was conducted and published the results provide insight into the scale of the problem [12]. 

Two aspects of the methodology of BET are of fundamental importance: (1) interpretation of the wave’s arrival time and (2) selection of the wave’s frequency. First of all, adequately detecting the time travel (*t*) is one of the most important obstacles in data interpreting of BET. Many studies were made considering this issue and many testing and interpretation methods have been proposed so far. To diverse methodologies created over the years, there can be included the simplest methods built on the immediate observation of the wave traced and measurements of the time interval between starting points. For interpretation of the received signals, more elaborated techniques, supported by signal processing and spectrum analysis tools. can be also applied [13]. All the methods created a deal with “appropriate” criteria to select the arrival time. The initial classification of such methods appears in Arulnathan et al. [14]. Viana Da Fonseca et al. [13] updated this classification more recently including developed methods and presenting a combined framework taking advantage of both time and frequency domain interpretations. Time domain methods are direct measurements based on plots of electrical signals versus time, whereas the frequency domain methods involve analyzing the spectral breakdown of the signals and comparing phase shifts of the components [15]. Several researchers, like Greening and Nash [16], prefer the frequency domain methods because they potentially allow the automation of signal processing and avoid the difficulties associated with picking the first arrival. On the other hand, they are either unreliable or require considerable user intervention to provide a reasonable result [17]. It was found that there is no specialized technique with an adequate level of accuracy and reproducibility to be adopted as a standard. The time domain and frequency domain analysis applied in this work to estimate the travel time is presented in detail later in the article. Experimental Results and Discussion. 

Considering the determination of the exact travel time of shear wave between the transmitter and receiver, it is also necessary to raise the issue of the wave’s shape transmitted through the soil sample. Most early studies using BET generated a single square-wave pulse [13]. The problem with the square wave is that it is composed of a wide spectrum of frequencies [18]. From the received signal of the square wave alone, it is uncertain whether shear wave arrival is at the point of first deflection, the reversal point, or some other point. To reduce the degree of subjectivity in the interpretation, and to avoid the difficulty in interpreting the square wave response, Viaggiani and Atkinson [15] suggested using a sine pulse as the input signal. Sine-wave pulses have become more popular, as these have shown to primarily give more reliable time measurements [19].

The second analyzed methodological aspect is the choice of the wave frequency used in the study. Different frequencies give different results, and not all devices suitable for testing have a choice of frequencies. Unfortunately, due to the apparatus diversity, but also the diversity of the soil material, again, it is not possible to adopt a uniform approach. There is one very important and widely commented guideline in the literature, related to the useful frequency range [20], described in detail further in the manuscript (part 3). Nevertheless, it is worth mentioning that it is related to the useable frequency range [21]. It is a condition that the travel distance (*L_TT_*) to the wavelength (*λ*) should not be less than 2, in some cases and some types of soil 3 (reaching according to some authors even range. 

2 < *L_TT_*/λ < 9 [21]). It is all due to a strong near-field effect, which can distribute the received signal in a manner preventing correct interpretation. The recommendation of TC-29 [12] in this aspect is to perform research in a broader scope of frequency spectrum and analyze the results in terms of their consistency. The authors’ examples illustrating this issue are shown in part 3 of the article. 

Among the factors affecting the accuracy of BET are also sample geometry and its size [21]. Arroyo [22] stated that the sample size could produce (a) effects introduced by end rebounds which provoke interferences and signal overlap, and (b) effects due to the cylindrical boundary that produce and interfered with signal where each frequency travels at a different velocity especially when wavelengths are comparable with the size of the specimen. Rio [23] in his thesis stated that the best results are obtained for slenderness ratios greater than 2. Therefore, specimens with small diameters are more affected by reflections from lateral boundaries [21]. 

To conduct BET, BE transducers are plugged into both ends of the specimen [24]. Historically there has been concern over whether the installation of BEs may cause some degree of disturbance to the examined samples. Boonyatee et al. [25] in their study investigated the effects of BE installation on *V_S_* measurements. The effects were inspected by comparing t*V_S_* obtained before and after the receiver, BE is penetrated into the soil sample. The penetration tests, performed by varying the rate of penetration, size of the sample, and consolidation pressure, revealed that the installation of piezoelectric transducers generates almost no disturbance.

Wave velocity depends on various soil parameters, namely, confining stress, void ratio, moisture content, etc. In recent years, there has been growing interest in understanding the potential influence of grain-size characteristics on *V_S_* or *G_max_*, such as, e.g., the effect of particle size [26], the effect of gradation [27], and the effect of fines [28]. 

Cho et al. [29] conducted notable studies on the effect of grain shape, reporting a summary from the literature about natural and crushed sands and quantified shape parameters for these sands. A database of shape parameters was created using the reference shape chart proposed by Krumbein and Sloss [30]. One of the significant findings of their study [28] was that a decrease in the roundness of sand grains leads to a decrease in shear wave velocity or stiffness. It is worth noting that analyzed data cover a range of sands with different size distributions. A careful examination of the database showed that the coefficients of uniformity (*C_u_*) of these sands vary from as low as 1.4 to as high as 5.5.

Quick communication about the fundamental question as to whether shear wave velocity (*V_S_*) of sand is dependent on particle shape is a work by Liu and Yang [31]. The authors proved that a sand specimen with angular particles tends to exhibit higher *V_S_* or *G_max_* values than a sand specimen with rounded particles. 

One of the particular studies of the effect of particle-size distribution on the stiffness of sand using the resonant column technique has been conducted by Wichtmann and Triantafyllidis [32]. It has been demonstrated for a constant void ratio that in the investigated range (0.1 mm ≤ *d*_50_ ≤ 6 mm, 1.5 ≤ $C_{u}=\frac{d_{60}}{d_{10}}$ ≤ 8) the small strain shear modulus (*G_max_*) does not depend on the mean grain size (*d*_50_), but significantly decreases with an increased coefficient of uniformity (*C_u_*), where *d*_60_ and *d*_10_ represent the particle sizes that 60% and 10% of the sand mass are smaller than, respectively. This result is in agreement with that of Iwasaki and Tatsuoka [33], which was also derived from several *RC* tests on the sand. For poorly graded sands (*C_u_* < 1.8, 0.16 mm ≤ *d*_50_ ≤ 3.2 mm) without a fines content (i.e., no grains smaller than *d* = 0.074 mm), the values of *G_max_*(*e*) did not depend on *d*_50_. Furthermore, Iwasaki and Tatsuoka [33] could not observe a significant influence on the grain shape. Similar *G_max_* values were measured for sands with round, subangular, and angular grains. However, the works of Wichtmann and Triantafyllidis [32] and Liu et al. [27] raise concerns that the variation of *V_S_* reported by Cho et al. [29] may not be a true reflection of the shape effect, but rather be associated with varying gradation. 

The experimental work of Patel et al. [34], showing an appreciable size dependence of small-strain stiffness, does not agree with that of Witchmann and Triantafyllidis [32] and Iwasaki and Tatsuoka [33]. It does, however, appear to be consistent with that of Bartake and Singh [35], who performed BE tests on three dry samples of sand with similar gradation and found that the *G_max_* value increased as *d*_50_ of sand decreased. It is worth recalling the laboratory observations of Sharifipour et al. [36], where the BE technique was used to measure shear wave velocity but in glass beads. This work adds further uncertainty viz., for glass beads of three different nominal sizes (1.0, 2.0, and 3.0 mm) the authors obtained an opposite result to Patel et al. [34]. The value of *V_S_* increased with increasing particle size. 

Another interesting research work was done by Menq and Stokoe [37] who performed *RC* tests on natural river sand with different *d*_50_ and *C_u_* values without fines content. For each sand, three different initial densities (loose, medium dense, and dense) were studied. In contrast to Iwasaki and Tatsuoka [33], a slight increase in *G_max_* with increasing *d*_50_ was measured when void ratio (*e*) and mean effective stress (*p*) were kept constant. Furthermore, the curves of *G_max_(e)* were steeper for the coarse material. Then, the authors observed that *G_max_* increases with *C_u_* for constant relative density (*D_r_*). It should also be mentioned a study of Lontou and Nikolopoulou [38], where they showed a slight increase of *G_max_* with the mean grain size up to *d*_50_ = 1.8 mm. Significantly higher values for *d*_50_ > 1.8 mm may be influenced by the small specimen size (diameter of the specimen equal to 4.8 mm). 

However, it should be as well emphasized that the experimental data from BET in the literature always seem to indicate that particle size affects *G_max_*, although opposite trends were observed in the *G_max_* variation with grain size. On the other hand, still, the *RC* test data always seem to suggest that there is not any particle size effect. In this respect, a question appeared: Does the testing method have any effect on small-strain stiffness? Yang and Gu [26] tried to find the correct answer. For the fine glass beads, the BE measurements of *G_max_* are comparable to the *RC* measurements, with differences being less than about 10%. Both tests showed a trend that *G_max_* decreases slightly with *d*_50_, particularly for glass beads at a loose state.

In a more recent study, Altuhafi et al. [39] introduced a new shape parameter SAGI to collectively account for aspect ratio, convexity, and sphericity of sand grains and showed for a database of natural sand that *G_max_*, normalized for size and gradation, increases with SAGI. In the current context, whether shear wave velocity or the associated stiffness depends on the particle size distribution and/or grain shape remains still an open question [29,40,41].

In this paper, an experimental investigation is performed to study some aspects of proper selection and verification of the test conditions for BET methods of shear wave velocity (*V_S_*) determination. Moreover, for laboratory research Warsaw glacial quartz sand with different grain-size distribution was chosen. An extensive array of bender element tests in triaxial apparatus is performed on saturated compacted natural sands. The main test results are presented along with their interpretation and discussion. These results show the complex nature of determining *V_S_* using the BET method. Despite the increasing popularity of BET, considerable uncertainty remains in signal interpretation, and thus in the estimated shear wave velocity and shear stiffness [26]. A good example of large scatter in *V_S_* results evaluated from a single BE test on a specimen of natural clay is shown in the work of Clayton [42]. Similar results were obtained from international parallel bender element tests on uniform Toyoura sand [43]. These observations, together with the contradictory results in the literature for stiffness variation with particle size, underline the need for a careful examination of BET, especially for granular material, in which wave propagation is complex owing to its particulate nature. Therefore, the authors made some efforts to clarify several issues closely related to the reliability of BE measurements:the characteristics of received signals in both the time and frequency domains over a wide range of excitation frequencies and wavelengths;how changes of particle size alter the characteristics of received signals;the performance of different interpretation methods under a variety of combinations of test; andconditions (i.e., grain size, excitation frequency, and confining stress).

## 2. Materials and Methods 

### 2.1. Characterization of Materials Used

To explore the influence of grain-size distribution on the shear wave velocity of saturated compacted sand, several series of experiments were performed on twelve different soil samples. These samples were made of fractionated glacial quartz sand with particle sizes ranging from 100 to 4000 µm. The research material was obtained from the Vistula river valley from the area of Warsaw. To receive the material with different grain size, i.e., medium-grained (coefficient of uniformity, *C_u_*, should be in the range from 6 to 15) and well grained (*C_u_* should be bigger than 15) [44], sand samples with the coefficient of uniformity of 12.0, 14.0, and 16.0 were prepared. Moreover, it was assumed that some of the tested soils, on the one hand, will compact well and be suitable, e.g., construction of embankments (coefficient of curvature, $C_{c}=\frac{d_{30}^{2}}{d_{10}\cdot d_{60}}$, should be in the range from 1 to 3). On the other hand, some of the samples will not be susceptible to compaction and therefore of little use for construction purposes (*C_c_* should not be between 1 and 3) [45]. To meet these requirements, sand samples with *C_c_* equal to successively 1.0, 2.0, 3.0, and 4.0 were prepared.

The method of organizing samples for laboratory tests included several phases. First, the sand was sieved through sieves with the following mesh sizes: 4.0, 2.0, 1.0, 0.5, 0.25, 0.125, and 0.063 mm (7 gradations). Then, the grain size distribution curves shown in Figure 1 were mixed from these gradations in the correct proportions to receive the samples planned. Following Eurocode 7 [46], the material was classified as coarse sand (CSa). The sands S1 to S4 (Figure 1a) have the uniformity coefficient of *C_u_* = 12.0, the effective particle size of *d*_10_ = 0.063 mm, the equivalent diameter of particles *d*_60_ = 0.760 mm, different mean grain sizes in the range 0.50 ≤ *d*_50_ ≤ 0.70 mm, and different curvature coefficient in the range 1.0 ≤ *C_c_* ≤ 4.0. The materials S5 to S8 (Figure 1b) have the uniformity coefficient of *C_u_* = 14.0, the effective particle size of *d*_10_ = 0.063 mm, the equivalent diameter of particles *d*_60_ = 0.885 mm, different mean grain sizes in the range 0.62 ≤ *d*_50_ ≤ 0.72 mm, and different curvature coefficient in the range 1.0 ≤ *C_c_* ≤ 4.0. The sands S9 to S12 (Figure 1c) have the uniformity coefficient of *C_u_* = 16.0, the effective particle size of *d*_10_ = 0.063 mm, the equivalent diameter of particles *d*_60_ = 1.0 mm, different mean grain sizes in the range 0.70 ≤ *d*_50_ ≤ 0.90 mm, and different curvature coefficient in the range 1.0 ≤ *C_c_* ≤ 4.0.

The basic properties of the test materials are summarized in Table 1. These are data on equivalent particle sizes: *d*_10_, *d*_30_, *d*_50_, *d*_60_, the minimum and the maximum void ratios (e_min_, e_max_) (determined according to Polish standard code [47], and the optimum moisture content (m_opt_) and the maximum dry density (ρ_dmax_) (from Proctor test).

According to literature-empiric diagrams [30] and after observing the individual particles of representative samples through a microscope (magnification 20 times), the natural glacial quartz sands from Warsaw comprise sub-rounded to particles. Images of the individual particles of representative samples are given in Figure 2. The shape of the particles in terms of roundness and sphericity [28] were both estimated to range from 0.5 to 0.7.

### 2.2. Experimental Equipment, Specimens Preparation, and Testing Program

In this study, shear wave velocity was directly measured for sand specimens using piezoelectric bender elements (the GDS Bender Element system) installed in the modified triaxial apparatus. The set-up of the apparatus, manufactured by the British company GDS Instruments Ltd. (Hook, Hampshire, UK), is shown in Figure 3. All the samples had a cylindrical shape with the same diameters and heights. The apparatus can accommodate a soil specimen 70 mm in diameter and 140 mm high, with a water-filled cell pressure and an internal linear variable differential transducer of high resolution. Diameter D = 70 mm is the maximum achievable diameter for which our specimens can be easily molded. The size and proportions of all examined specimens were in agreement with the suggestions of previously reported works (see Introduction). The modified triaxial apparatus is mounted with vertically placed piezo-element inserts, produced by the aforementioned company GDS Instruments Ltd., on the top cap and the pedestal, respectively, which work as both bender elements and extender elements, similar to the configuration described by Leong et al. [48]. This allows the propagation of S-waves (bender element configuration) and P-waves (extender element configuration) through the body of the specimen. Consequently, measurements of the shear and compression wave velocities (*V_S_* and V_P_, respectively) can be carried out based on the obtained wave arrival times with the wave propagation direction along the specimen axis. 

The length of the bender element inserted into the soil specimen was optimized (always about 1.5 mm) to avoid compromising the power transmitted or received by the elements. This is achieved by fixing the element further down inside the insert and then filling the remaining volume with flexible material. This allows the element to achieve maximum flexure at its tip, while only protruding into the sample by a reasonable distance. Advantages of this include prolonged life by increased resilience to breakage and easier sample preparation, particularly on very stiff samples where only a small recess for the element is required.

When performing BET, one of its most important aspects is the phase orientation of the elements [24]. The authors always checked the relationship between the received signals concerning the source signal. The desirable orientation was “n-phase”. If the orientation was correct, the source and received traces were exactly “in-line”.

During BET, shear wave velocity was calculated from the simple measurement of propagation distance (Δs) and propagation time (Δt). Based on several previous works, it is generally accepted that the travel distance is the distance between the tips of two *BE*s [21]. Therefore, in the presented research the travel distance was the tip-to-tip length, which is the height of the specimen minus the length of each *BE*. 

Specimens of Warsaw glacial quartz sand were prepared using the moist tamping method by under compaction technique, which is similar in principle to the methods used by several researchers in testing granular soils (see, e.g., in [49]). The optimum moisture content was adopted to receive moist specimens. A predetermined mass of soil was mixed with an appropriate amount of water to obtain a moisture content close to the optimal one. Each soil specimen was prepared in five layers, and it was compacted using a tamper after placing each layer [27]. No obvious segregation was observed during specimen preparation. After sample preparation, the drive head and the load cell were installed. To stabilize the specimens, a suction of 35 kPa was applied to the specimens. The dimensions of the specimens were measured accurately and the initial void ratio was determined. The cell and back pressure were then increased simultaneously to keep a constant isotropic effective stress of 35 kPa.

When a triaxial cell was assembled a specimen was flushed with CO_2_ and then with de-aired water. The specimens were saturated with a Skempton’s pore-water pressure parameters B-value over 95%. Subsequently, to reach the target, the confining and backpressure were raised step by step with a difference of 30 kPa. All the specimens were tested in a fully saturated state. After saturation, they were subjected to an isotropic stress path (compression), and the dynamic tests (S-wave and P-wave measurements by bender elements) were conducted at mean effective stress *p*′ = 45 kPa for specimens S1–S10, except for specimens S11 and S12. These were tested at progressively increasing mean effective stress equal to *p*′ = 45, 90, and 180 kPa. During each consolidation process, the volume changes of the specimen and the corresponding deformation were measured. In the next step, after the completed consolidation, wave velocities measurements were performed under a range of systematically changing excitation frequencies. The frequency range was selected based on the literature review and the authors’ own experience. This article focuses primarily on shear wave velocity using one type of signal input: sinusoidal signal. However, for most of the specimens, compression wave velocity was also estimated. Note that specimens S1–S10 were tested in one series, at one confining stress (45 kPa). In contrast, specimens S11 and S12 were subjected to multi-stage consolidation, each one with increasing confining stress. In Table 2, the summary of the test series is presented.

## 3. Experimental Results and Discussion

### 3.1. Synopsis of Experimental Results

#### 3.1.1. Piezo-Elements Signal Analysis

The estimation of the wave velocities from the bender/extender elements was performed primarily by adopting two different approaches related to the observation of the source and received bender/extender element signals, namely, the first time of arrival method (FTA; a different name for start-to-start method) and the peak-to-peak method (PTP). They are both considered as typical techniques applied in the time-domain (TD). The time-domain methods generally determine the travel time directly from the time lag between the transmitted and received signals [24,43,50]. 

The first above-mentioned method is based on the visual inspection of the received signal and is still quite controversial and subjective due to the complex received signal, wave’s reflection, and the near-field effect. Many studies reported that the near-field effect decreases with the increase of frequency or the ratio of wave path length to wavelength ($\frac{L_{tt}}{\lambda })$. Arulnathan et al. [14] reported that the near field effect disappears when this ratio is larger than 1.0. Pennington et al. [51] pointed out that when the $\frac{L_{tt}}{\lambda }$ values range from 2.0 to 10.0, a good signal can be obtained. Wang et al. [52] advocated a ratio greater than or equal to 2.0 to avoid the near field effect. Similarly, a value of 3.33 was recommended by Leong et al. [53] to improve the signal interpretation.

The peak-to-peak method is also widely applied in signal interpretation. In this technique, the time delay between the peak of the transmitted signal and the first major peak of the received signal is regarded as the travel time [54]. As the frequency of the received signal may be slightly different from that of the transmitted signal, and the nature of the soil and size of the sample often affect the shape of the signal which could present more than one peak, great attention should be paid to the calculation of travel time using this method. 

Typical plots of the input and output waves from bender and extender element tests for the specimens with code name S11 and S12 are given in Figure 4a,b. A wave was produced by a displacement in the source transducer due to applied excitation voltage equal to 10 V or 14 V. Wave transmission created a displacement in the receiver, which in turn resulted in a voltage that could be measured. The frequency of the input signal was variable and decreased from the value *f* = 25 kHz to *f* = 2.0 kHz for S-wave velocity and from *f* = 100 kHz to *f* = 10 kHz for P-wave velocity. For each test frequency, five separate source element triggers were applied to the specimen, with the received signal output then stacked in the time domain to remove random signal noise [55].

From the data shown in Figure 4a, it can be easily seen that the maximum and first peaks do not coincide for tested sandy soils. These oscillations are derived from ambiguous peaks observed at the beginning of the received signal. This behavior [43] is due to reflected P-waves and does not represent the arrival of an S-wave [55]. The effect of the reflected wave is more pronounced for sandy soils than for any other kind of soil because of long reverberations and low damping. At this place, there is also a phenomenon called the near-field effect observed. The near-field component of the wave causes the transmitted wave to be distorted at its origin point and the wave starts with downward or upward deflection [56]. To minimize the near-field effect, as suggested in the literature [57,58], the higher input frequency and wavelength were used. However, in some examined cases, despite the triggering frequency higher than 2 kHz, the $\frac{L_{tt}}{\lambda }$ ratio was around 1 or even slightly less. At that time, the near-field effect was not avoided. 

A comparison between the two selected different interpretation methods of shear and compression wave velocities is given in Figure 5. The data shown in this figure are clustered around the 45^°^ lines, indicating an excellent agreement between the two analyzed methods. For more than half the number of the specimens (around 55% of the test results), higher shear wave velocity values were obtained from *PTP* method. The situation is similar in the case of compression wave velocity values. Here, for 61% of the data, *V_P_* from the *PTP* method is greater than *V_P_* from the *FTA* technique. Nevertheless, the differences between the results from both compared methods, for 99% of the results, are up to 10% for S-wave, and up to 20% for P-wave. The minimum difference between studied techniques in the case of *V_S_* is 0.1 m⋅s^−1^, whereas the maximum is −42.1 m⋅s^−1^. In the case of *V_P_*, the minimum difference is equal to 2.0 m⋅s^−1^, whereas the maximum is –223.0 m⋅s^−1^. From Figure 5 can be also seen that the *FTA* method of identification results in a relatively smaller variation as compared with *PTP* method. 

In Figure 6, shear wave velocities obtained for different source wave frequency for the test specimens S11 and S12 are presented. In this exemplary figure, a clear scatter of the results of *V_S_* calculated based on two selected signal analysis methods depending on the source frequency can be perceived. In the case of the lowest frequencies used (*f* < 3.3 kHz), the significant variation of the results was obtained (from around 4 m⋅s^−1^ to 30 m⋅s^−1^ —specimen S11 and from around 11 m⋅s^−1^ to again around 30 m⋅s^−1^ —specimen S12). It is most likely related to the aforementioned phenomenon of the near field effect. This was explained, e.g., in the work of Sánchez-Salinero et al. [59]. The authors suggested the lower limit of frequency of 3.3 kHz as the value which corresponds to an approximate propagation distance-to-wavelength ratio equal to two, avoiding data that may include near field effects. In the case of specimen S11, a relatively large discrepancy between the S-wave velocities was noted for *f* = 5 kHz (average 14.5 m⋅s^−1^), whereas for specimen S12, the scatter in estimated *V_S_* values for this frequency as well for frequency *f* = 10 kHz was the minimal (average 0.1 m⋅s^−1^). Therefore, for the investigated soils, it seems appropriate to choose a frequency closest to 10 kHz as the characteristic frequency. Generally, for all tested Warsaw glacial quartz sands, as the source frequency increases from the characteristic frequency, comparable *V_S_* values can be received, regardless of the travel time determination technique adopted (here, *FTA* and *PTP*). 

In Figure 6, the variations in shear wave velocity values for different mean effective stress (*p*′ = 45, 90, and 180 kPa) are shown beside. Shear wave velocities increase as the effective stress increases, as expected. On average, this increase for all obtained data is 16–17%.

To verify any possible influence of multi-stage consolidation process on the *V_S_* results, shear wave velocities of specimen S12 from three consecutive consolidation stage, for mean effective stress *p*′ = 180 kPa, corresponding to the test series, with numbers respectively XIV, XV, and XVI, are summarized in Figure 7. Based on the analysis of the results, it was found that the differences between the respective *V_S_* values from individual test series of the same soil specimen were on average 4 m⋅s^−1^ (around 1%). The multi-stage consolidation procedure seems to be in fairly good agreement with the conventional one.

#### 3.1.2. Multi-Method Automated Tool for Travel Time Analyses—GDS BEAT

A lack of agreement when interpreting the S-wave travel time from the bender elements test triggered the development, by the British company GDS Instruments, of a new software tool to automate the interpretation process using several analysis methods recommended in the literature [12,14,60]. The main aim of this tool was to allow travel time estimations to be conducted objectively via a simple user interface, providing both visual and numerical representations of the estimated travel times. Implementation of the tool was completed by creating Add-Ins for Microsoft Excel: The Interactive Analysis tool and the Batch Analysis tool, a decision based on the ubiquitous use of the software. The details of the GDS Bender Element Analysis Tool user interface can be found in the work of Rees et al [55].

For implementation, variations of the three different approaches were chosen:observation of points of interest within the received wave signal via software algorithm (time-domain technique),cross-correlation of the source and received signals (time-domain technique), anda cross-power spectrum calculation of the signals (frequency-domain method).The operation of the program is mainly based on one specific method of numerical analysis of the obtained results, using three factors:objective marking of points A, B, C, D (Figure 8) with the help of a software algorithm;mutual connection of generating and receiving elements signals; andcalculation of the signal power curve spectrum for time estimation in the frequency-domain method.

The major first peak (point D) is located by scanning the received signal and determining the maximum, as well as the most positive output. The corresponding time signature defines point D. Next, point B is defined by scanning the wave signal from time zero up to point D and locating the minimum, as well as the most negative output. The time signature corresponding to this minimum thus describes point B. To find point C, it is required to scan the received wave signal between points B and D in such a way as to locate the output closest to zero. The corresponding time on the timeline is the one that is assigned to point C, whereas point A is computed during the interaction of the individual values. Starting at time zero, the mean and standard deviation of 10 consecutive outputs (e.g., *n*_1_–*n*_10_) are calculated, followed by 5 consecutive outputs (*n*_11_–*n*_15_). Subsequently, it is judged whether at least 3 standard deviations are more negative than the calculated mean. If it is true, the time signature of the first of the 5 subsequent outputs (i.e., *n*_11_) is used to define point A. If it is false, the iteration proceeds by determining the mean and standard deviation of the next set of 10 consecutive outputs (i.e., *n*_2_–*n*_11_) until a “true” condition are reached [55]. 

It is interesting how within BEAT the cross-correlation and the cross-power spectrum calculation of source and received wave signals work. Cross-correlation values are calculated from the source and received element signals at each data time stamp. The time at which the maximum calculated cross-correlation value occurs is then used as an estimate of the shear wave travel time. The cross-power spectrums, obtained via a Fast Fourier Transform (FFT), are used to create a phase angle versus frequency plot. The slope of this plot is then used to estimate the shear wave travel time, based on a linear best fit across a defined frequency window. Note, GDS BEAT automatically uses a frequency window of 0.8 to 1.2 times the specified source element frequency; however, this can be manually altered by the user when running the Interactive Analysis. 

In Figure 9 and Figure 10, a summary of the *V_S_* values obtained by various techniques of travel time determination is presented. The data combined in Figure 9 concern the first ten specimens, namely, from S1 to S10, tested at *p*′ = 45 kPa. Those in Figure 10, however, relate to two specimens, S11 and S12, isotropically consolidated to three mean effective stresses, *p*′ = 45, 90, and 180 kPa, and additionally subjected to multi-stage consolidation. All the results are resumed here, regardless of the value of the $\frac{L_{tt}}{\lambda }$ ratio. By performing the GDS BEAT program, the results concerning the following four automated analysis methods were received: three time-domain techniques, namely, peak-to-peak, zero-crossing, and cross-correlation, and one frequency domain technique, i.e., cross-spectrum. These four methods were subsequently compared with two non-automated subjective analyses, including the first time of arrival method and the peak-to-peak technique (discussed in 3.1.1. Piezo-elements signal analysis). From both Figure 9 and Figure 10, it can be noted that the *V_S_* values are nonuniform for different source frequencies. 

The values of *V_S_* in Figure 9 indicate significant variation in the cost estimates obtained using the cross-spectrum calculation. The dispersion of the results for this method ranges from 15.8 m⋅s^−1^ (specimen S6) to 76.2 m⋅s^−1^ (specimen S1). The scatter observed from the zero-crossing method is also noteworthy, especially in the case of two specimen S5 (Δ*V_S_* = 124.4 m⋅s^−1^) and S6 (Δ*V_S_* = 236.4 m⋅s^−1^). For the rest of the tested specimens, however, these are the values at the level of around 20.0 m⋅s^−1^. Conversely, the difference in the results obtained from the peak-to-peak automated analysis made by BEAT (±3.7 m⋅s^−1^) and the cross-correlation function (±5.0 m⋅s^−1^) is relatively minimal. The *V_S_* results in Figure 9 show also that in the case of Warsaw glacial quartz sand tested at the preset pressure of 45 kPa the smallest dispersion of shear wave velocity (±16.0 m⋅s^−1^) was recorded for the smallest source frequencies, i.e., *f* = 2.2 kHz and *f* = 2.5 kHz. 

Inspection of Figure 10 indicates that the largest scatter of the results was obtained again for the cross-spectrum calculation, ranging from 193.3 m⋅s^−1^ (specimen S12/XV-2) to 593.2 m⋅s^−1^ (specimen S12/XIV-2). Analyzing the other methods, the zero-crossing technique is also characterized by a quite large discrepancy in the *V_S_* values, where Δ*V_S,min_* = 23.8 m⋅s^−1^ and Δ*V_S,max_* = 356.3 m⋅s^−1^. It may be suggested that estimates taken from these two above mentioned methods may be unreliable. In particular, the scatter observed from the cross-spectrum analyses has been previously reported following other studies [49]. Furthermore, the authors of the article themselves in their research devoted shear wave velocity of two types of anthropogenic material [42] received quite a large scatter in the submitted data from the frequency domain method. Therefore, it is quite dangerous to identify the arrival time only with this method. A significant decrease in the discrepancy of the results (±15.0 m⋅s^−1^), for specimens S11 and S12, was obtained for the peak-to-peak method specified by the user. The scatter in calculations from the cross-correlation function for these two specimens was also rather small (±40.0 m⋅s^−1^). These values suggest each method is relatively robust, an observation also made for the cross-correlation function after reviewing recent studies comparing analysis methods [15]. In the case of specimens S11 and S12, tested at three preset pressures (45, 90, and 180 kPa), the smallest dispersion of shear wave velocity (±131.0 m⋅s^−1^) was recorded for the source frequencies equal to *f* = 3.3, 4.0, and 5.0 kHz. The highest scatter of the results were obtained in the case of both the lowest (*f* = 2 kHz) and the highest (*f* = 25 kHz) frequencies set in BET. 

In Figure 11, the mean values of *V_S_* calculated for all methods of interpretation listed in the article, after applying the frequency criterion, are presented. Therefore, wave velocities from the frequency of 3.3 kHz were used for further discussion of the results, avoiding data that may include near-field effects [59]. For most of the tests of Warsaw glacial quartz sands (for 99% sand specimens), the lowest average values of *V_S_* were obtained from the peak-to-peak methods produced by BEAT. The highest *V_S_* values for 11 specimens were gained from the frequency domain technique, whereas the remaining 9 specimens had their highest values from the zero-crossing method. 

In the next order, the statistical analysis of all the data presented in Figure 11 was executed. The use of the frequency criterion did not significantly affect the obtained results. The notable variation in the mean *V_S_* values resulted again in applying the cross-spectrum technique (±775.6 m⋅s^−1^). A small spread of the results was once more provided by the peak-to-peak method specified by the user (±107.7 m⋅s^−1^) and the cross-correlation function (±108.0 m⋅s^−1^). The smallest one, however, was obtained using the first time of arrival method (±103.8 m⋅s^−1^). This technique of travel time identification in BEs testing also gave the lowest standard deviation, amounting to 31.8 m⋅s^−1^. After estimating standard error, for the first time of arrival method and the cross-correlation method, the standard error is insignificant: just 1.2 m⋅s^−1^. This can mean that these two above analyzed interpretation methods are correct. An analysis of uncertainty was also required, to approve the precision and credibility of this study. The relative uncertainty in the range of 0.1% to 10% is typical for laboratory experiments [42]. Based on the data summarized in Figure 11, lower uncertainty of the results were gained for the time domain techniques, at the level of 16% (the first time of arrival method) and of 18% (the cross-correlation method and the peak-to-peak specified by user).

In this study, the comparison of four time-domain methods, namely, the user-specified peak-to-peak together with the peak-to-peak by BEAT (Figure 12), as well as the user-specified first time of arrival together with the zero-crossing (Figure 13), was completed. For each pair, the same methods are analyzed, but the first one is the subjective analysis made by the authors themselves during BE tests, whereas the second one is the automated analysis by the performance of BEAT. The minimum and maximum differences in the shear wave velocity values are included in the figures below. It is visible that these differences depend on the tested sand specimens. The average difference between the two peak-to-peak techniques was around 22.2 m⋅s^−1^, which is 11%. The average difference between the first time of arrival and zero-crossing techniques was around 33.4 m⋅s^−1^, which is 14%. The results obtained can be considered as preliminary results suggesting the use of BEAT may decrease subjectivity when interpreting travel times using standard observational techniques, while still allowing accurate estimates of the shear wave velocity values, keeping in mind the type of soil tested.

For further analysis, i.e., the influence of grain size characteristics on shear wave velocity of Warsaw glacial quartz sand, the mean values of *V_S_* obtained for *p*′ = 45 kPa were chosen. Moreover, such results were selected for which the quality of received signals was found to be satisfactory, also near-field effects and attenuation were found to be reduced. Then, the focus was put only on two interpreting travel times method: on the standard observational technique—the first time of arrival method (FTA)—and on the cross-correlation (CC) function obtained via BEAT analyses. These are the methods that provide the most consistent results for the studied soils. 

### 3.2. Effect of Grain Size Characteristics

In Figure 14, the values of mean shear wave velocity obtained for all tested Warsaw glacial quartz sands at *p*′ = 45 kPa, from two selected time-domain methods for determining the wave travel time, versus the coefficient of curvature, are presented. This parameter took values from 1 to 4, whereas the test material was divided into three groups depending on the coefficient of uniformity. No clear dependence of the curvature coefficient (*C_C_*) on the *V_S_* values can be detected from the date in Figure 14. The highest values of *V_S_* have examined sands characterized by *C_C_* = 2, regardless of the *C_U_* value. The smallest shear wave velocity was obtained for sands with *C_C_* = 3. The greatest scatter of the *V_S_* results characterizes the specimens with *C_C_* = 4 (Δ*V_S_*_,avg_ = 10%) and *C_C_* = 2 (Δ*V_S_*_,avg_ = 8%), in the case of the *FTA* method. The smallest, however, have the specimens with *C_C_* = 1. When considering the results from the *CC* method, the significant variation in the *V_S_* estimates, i.e., 10%, was gained for sands with *C_C_* = 1, while the minimal one for sands with *C_C_* =3 (Δ*V_S_*_,avg_ = 5%).

In Figure 15, it is demonstrated that, in contrast to the curvature coefficient, the shear wave velocity of Warsaw glacial quartz sand is influenced by the uniformity coefficient (*C_U_*). As *C_U_* increased, shear wave velocity increased too. The only doubts can be raised by the results of sands with *C_C_* = 1. In the case of the *FTA* method, the greatest scatter of the *V_S_* results characterizes the very well-graded specimens, with *C_U_* = 16 (Δ*V_S_*_,avg_ = 16%), while the smallest one characterizes the well-graded sands with *C_U_* = 14 (Δ*V_S_*_,avg_ = 12%). The dispersion of the *V_S_* values is more significant for the second considered method. Here, namely, sands with the lowest *C_U_* are characterized by the greatest variability of *V_S_* (Δ*V_S_*_,avg_ = 18%). On the other hand, for the specimens with *C_U_* = 14, the scatter of the results is relatively minimal (Δ*V_S_*_,avg_ = 5%).

In Figure 16, the mean shear wave velocity is plotted versus the mean grain size. The effect of *d*_50_ on the *V_S_* values is hardly demonstrated in this study. In some of the tested materials, i.e., when *C_U_* = 12, *V_S_* decreased around 16% with increasing *d*_50_. However, for Warsaw sands with *C_U_* = 14 or *C_U_* = 16, as *d*_50_ increased, an average of 9% to 15% increase in the *V_S_* results was noted. This decrease in the *V_S_* values is more visible than the increase.

## 4. Concluding Remarks

The results of 12 clean quartz samples of sands from the Warsaw region presented in the article show, first of all, the complex specificity of performing shear wave velocity determination using the BET method. It is visible that despite the universality of this type of apparatus, special attention should be paid to the methodology of work and the correct selection of the test parameters. As has been shown, both the aspect of selecting the frequency of the test and the method of interpreting the results appropriate for a given geomaterial are very important. Users of the BET method should be aware of all factors that may influence the obtained results.

Concerning the difference in the arrival time identification method for Warsaw glacial quartz sandy soils, the time domain interpretation methods, namely, the first time of arrival (FTA) and the cross-correlation (CC) techniques, provide *V_S_* results that are more consistent compared to the other methods. The scatter of the results for these two methods was undoubtedly smaller, even up to 7 times smaller than for the most questionable method, as it turned out here the cross-spectrum (CS). Quite small values of standard deviation and standard error allow us to conclude that these methods are relatively robust. The latter of these two techniques, i.e., *CC*, was used in the application of the GDS Bender Element Analysis Tool. The *CS* method proved to be the interpretative technique that must be used with great caution for the tested sandy samples. 

If data at different frequencies are available within the sine input wave, the results of shear wave velocity for analyzed soils by input frequency closest to 10 kHz should be selected. A frequency close to this value 10 kHz is a characteristic frequency for Warsaw glacial quartz sandy soils.

It should be noted as well that BEAT can offer an accurate, objective explanation of BET data via a simple user interface. By using such a tool, a significant reduction in the time needed for the interpretation of the *V_S_* results by different methods, at the same time, deserves a great emphasis. Automation is also a direction that allows objectification and popularization of the interpretation method.

A comprehensive experimental program has been performed using bender elements incorporated in the triaxial apparatus to define the combined effects of grain-size characteristics on the shear wave velocity of Warsaw glacial quartz sands too. As a measure of grain-size characteristics, three parameters were involved: the uniformity coefficient (C_U_), the curvature coefficient (*C_C_*), and the mean grain size (*d*_50_). Despite the narrow range of variability of the particle size curves, the test results show that shear wave velocity is not affected by both the *C_C_* and *d*_50_ of the tested material. In contrast, for most of the analyzed cases, the *V_S_* values significantly increased with increasing the uniformity coefficient, with an average increase of 13.5%.

In future work, the laboratory tests will be extended to include Warsaw quartz sands with different grain size distribution curves. For each new material, tests with different pressures and densities are planned. After database expansion, the authors would like to examine some selected from the literature expressions of *G_max_* to inspect if they can predict the measured values of *G_max_* with the right level of accuracy. Additionally, comparative tests are planned to actually compare the stiffness of Warsaw quartz sands from various laboratory methods: bender elements (BE), resonant column (RC), and torsional shear (TS).

## Figures and Tables

**Figure 1 materials-14-00544-f001:**
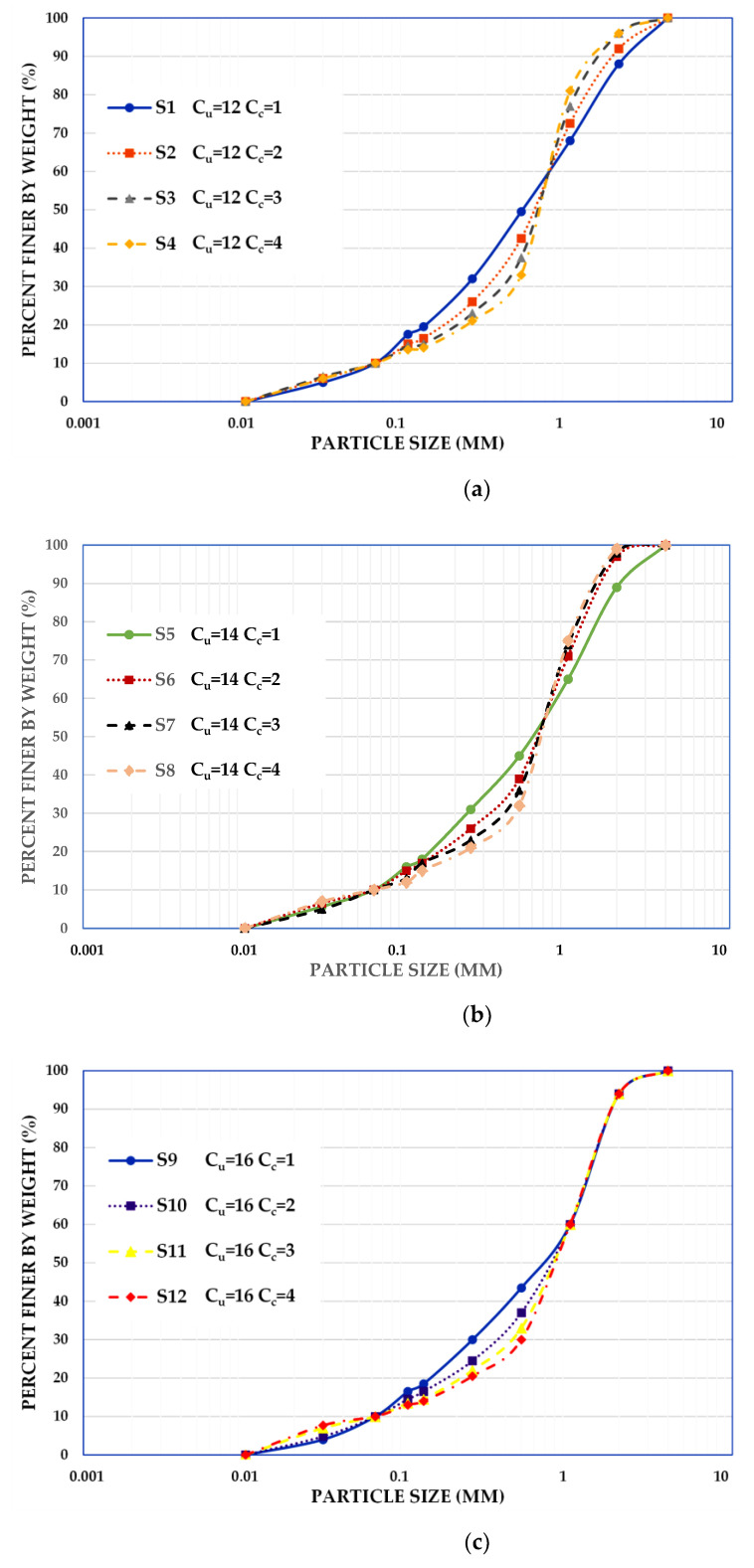
Tested grain size distribution curves for the group of sands with *C_u_* equal to (**a**) 12, (**b**) 14, and (**c**) 16.

**Figure 2 materials-14-00544-f002:**
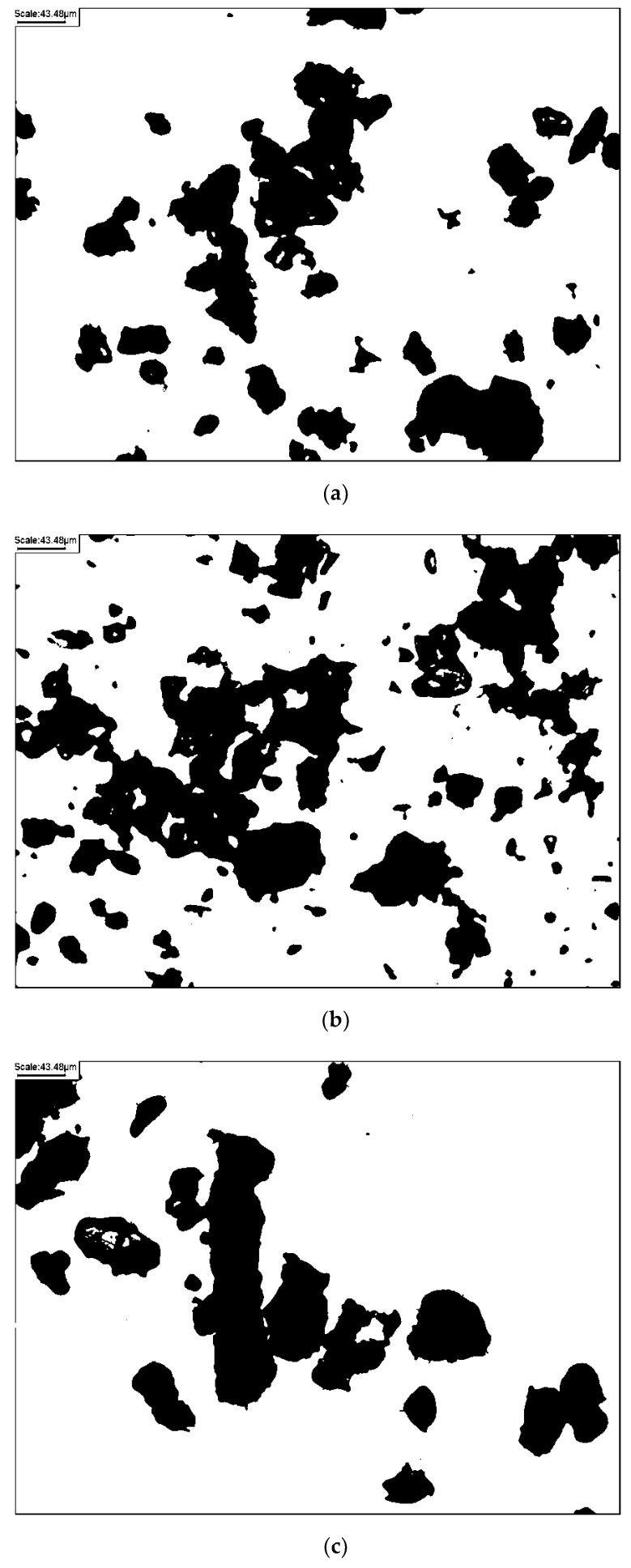
Microscopic images of individual particles of representative samples of this study: (**a**) S2, (**b**) S8, and (**c**) S9.

**Figure 3 materials-14-00544-f003:**
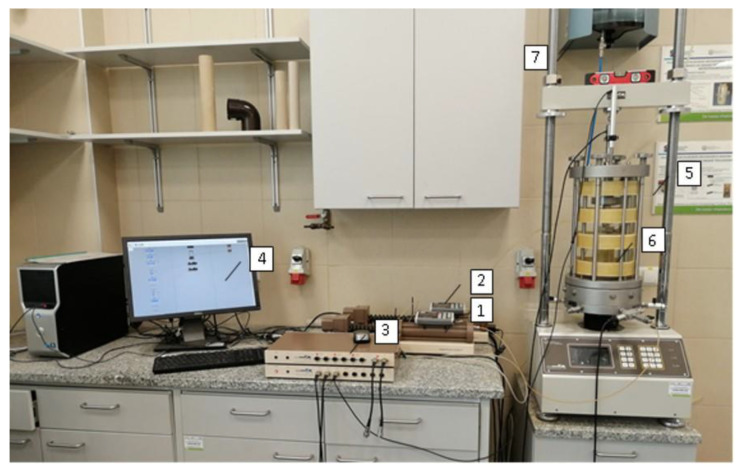
Set-up of dynamic testing system at Water Centre in Warsaw University of Life Sciences: (1) back pressure controller, (2) cell pressure controller, (3) data logger, (4) PC and control software, (5) frame, (6) cell, (7) water tank.

**Figure 4 materials-14-00544-f004:**
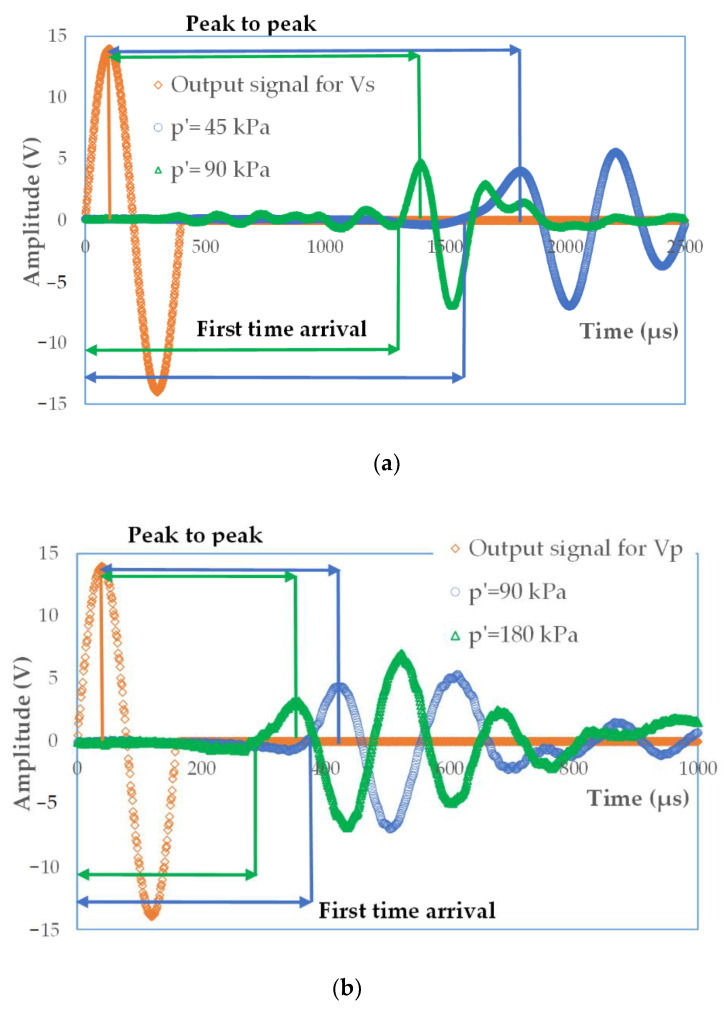
Example of the signal analysis/interpretation for (**a**) the bender element tests for specimen S11, *p*′ = 45 kPa, *f* = 10 kHz, and (**b**) the extender element tests for specimen S12, *p*′ = 90 kPa, *f* = 12.5 kHz.

**Figure 5 materials-14-00544-f005:**
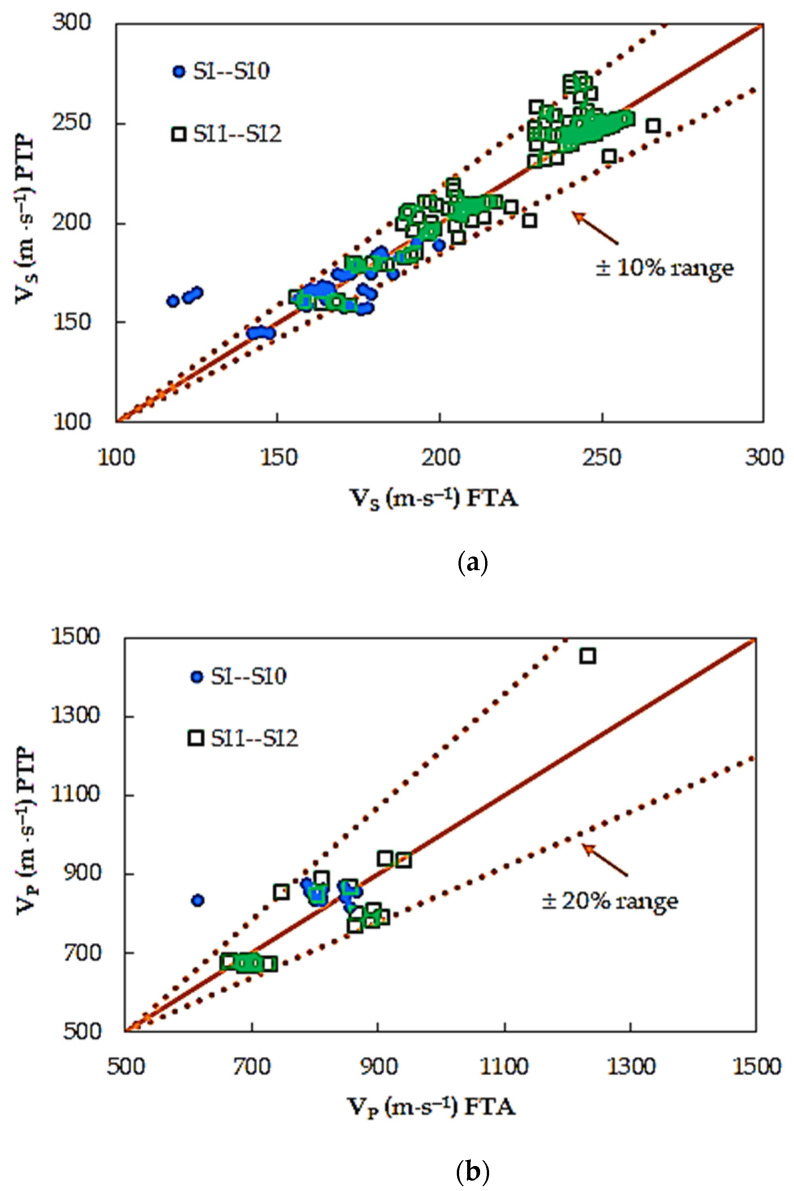
Comparison of (**a**) S-wave and (**b**) P-wave velocities, between the first time of arrival (FTA) and peak-to-peak (PTP) methods of signal analysis.

**Figure 6 materials-14-00544-f006:**
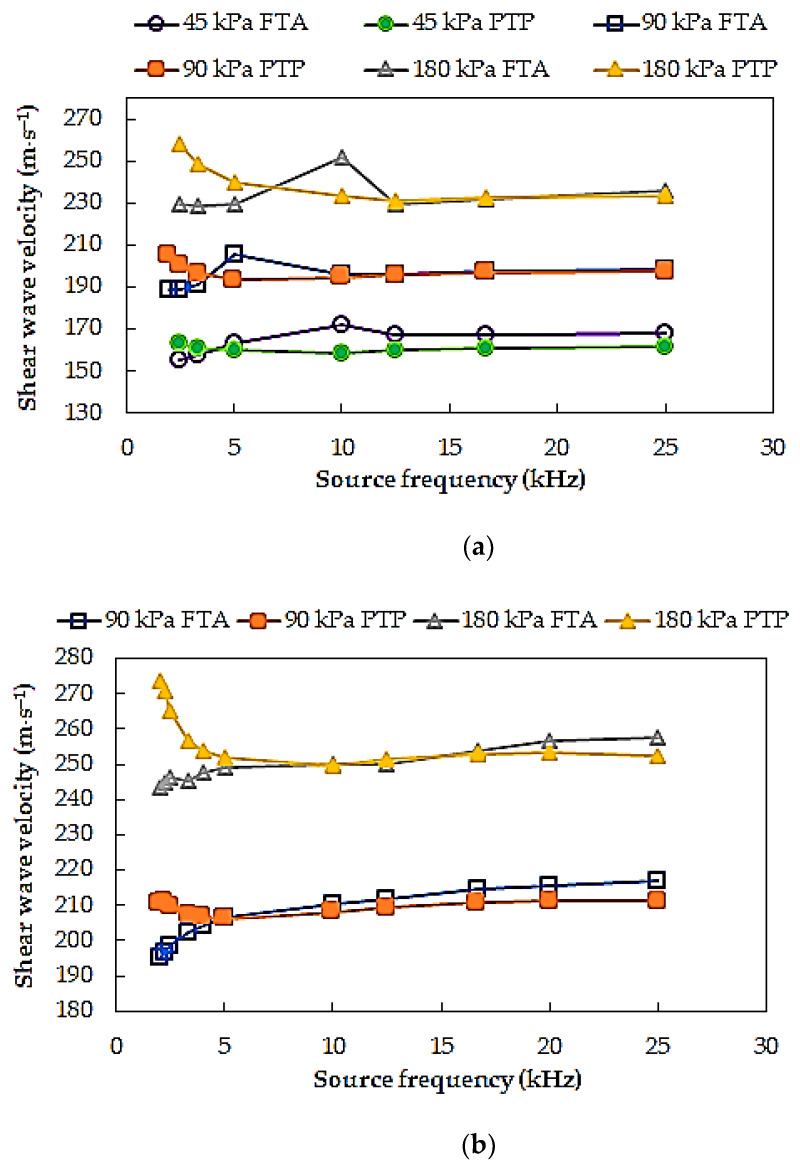
Shear wave velocities for different source frequencies and different mean effective stress, data for the test specimen: (**a**) S11, test series XI, steps 1, 2, 3, and (**b**) S12, test series XV, steps 1, 2.

**Figure 7 materials-14-00544-f007:**
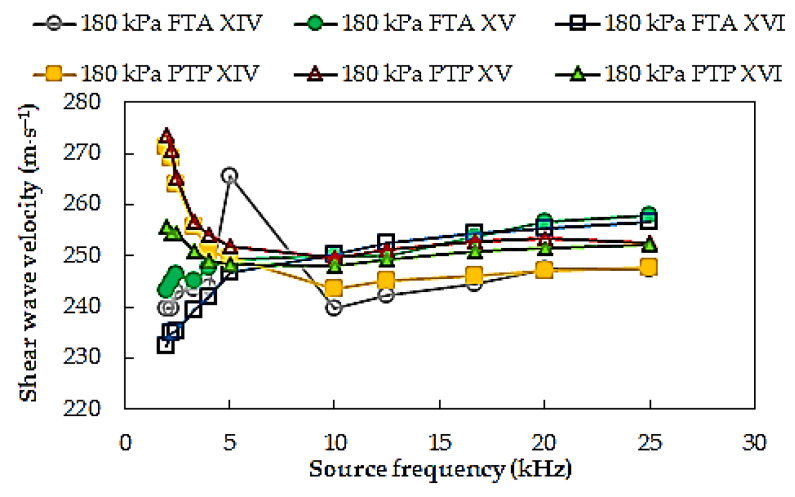
Shear wave velocities from multi-stage consolidation process.

**Figure 8 materials-14-00544-f008:**
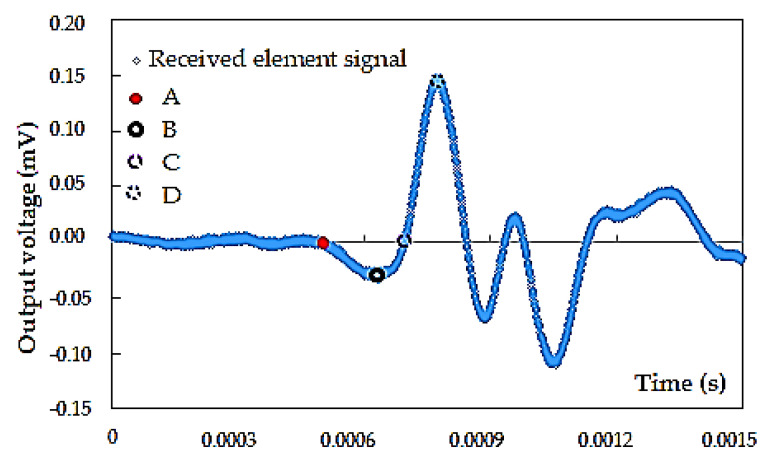
Receiver time charts: (**A**) the first deflection, (**B**) first bump, (**C**) zero-crossing, and (**D**) first major peak. An example of the test results from specimen S1, *f* = 4 kHz.

**Figure 9 materials-14-00544-f009:**
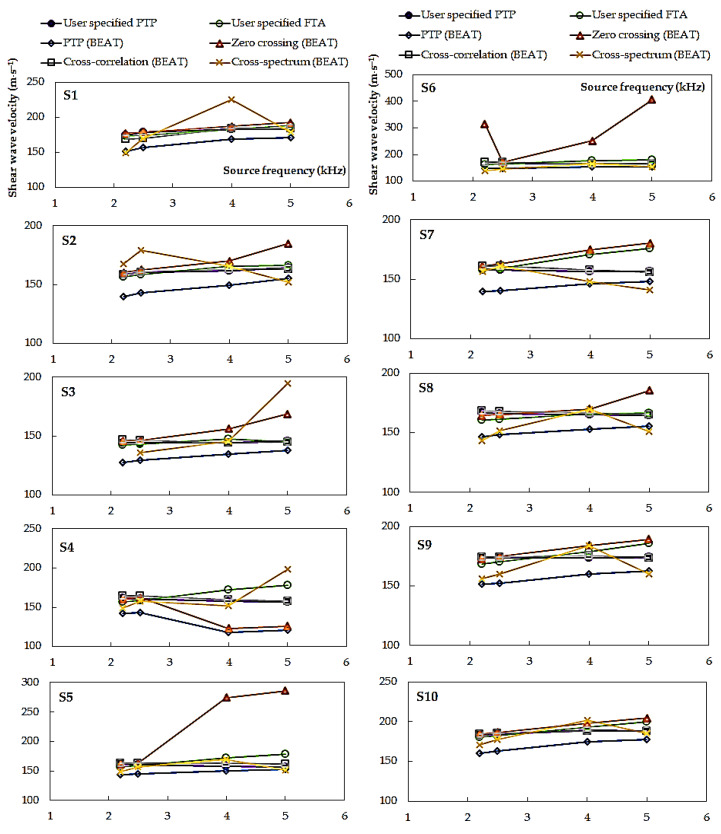
Shear wave velocities calculated from various methods of interpretation of *BE* test results, data for specimens S1–S10.

**Figure 10 materials-14-00544-f010:**
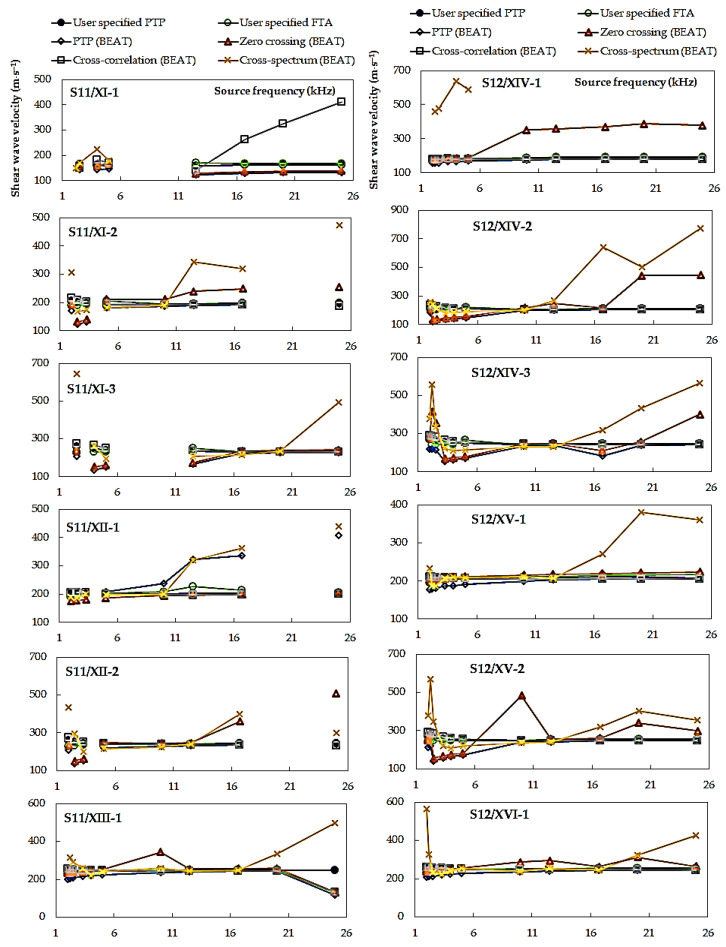
Shear wave velocities calculated from various methods of interpretation of BE test results, data for specimens S11–S12.

**Figure 11 materials-14-00544-f011:**
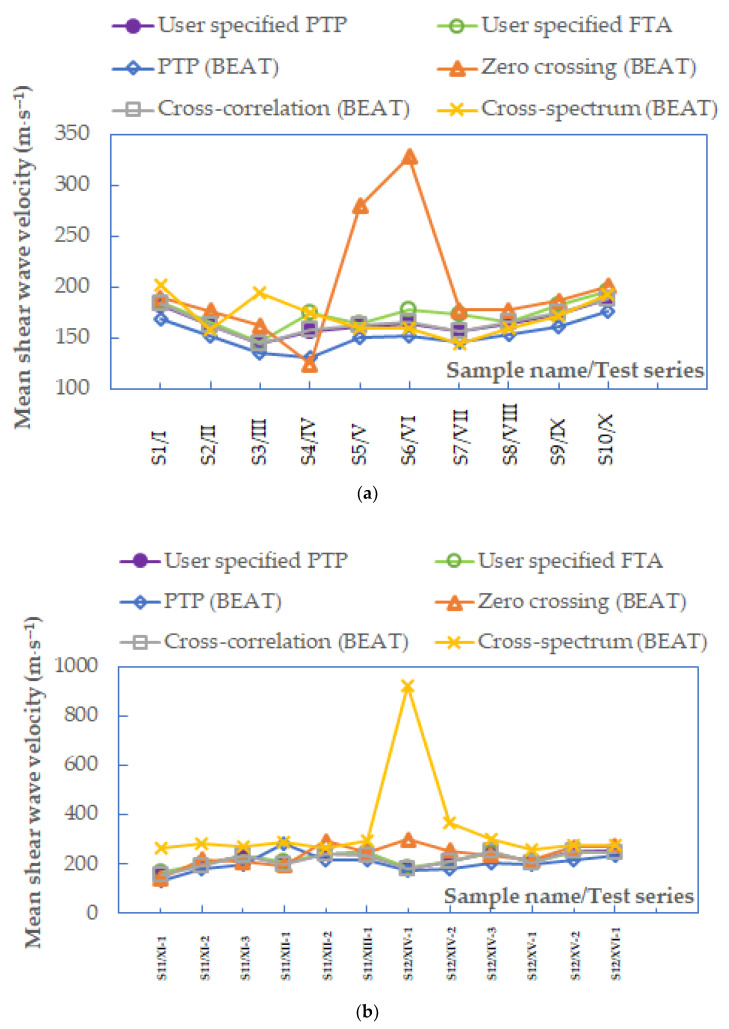
Summary of mean values of shear wave velocities from different interpretation methods when *f* < 3.3 kHz, data for specimens (**a**) S1–S10 and (**b**) S11–S12.

**Figure 12 materials-14-00544-f012:**
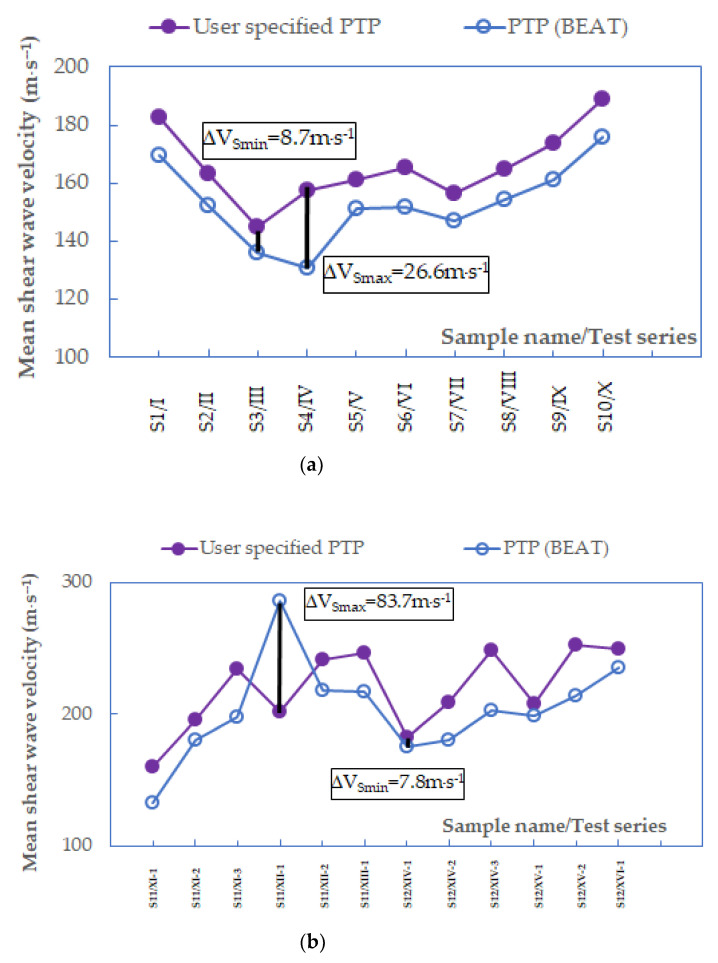
Mean values of shear wave velocities from peak-to-peak methods when *f* < 3.3 kHz, data for specimens (**a**) S1−S10 and (**b**) S11–S12.

**Figure 13 materials-14-00544-f013:**
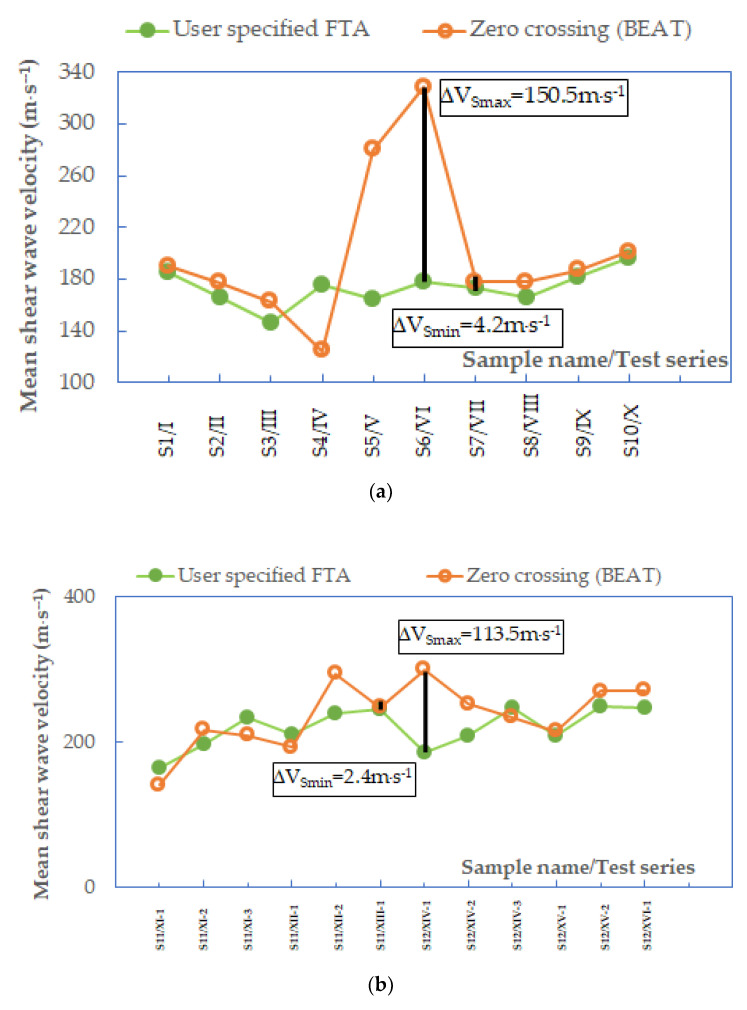
Mean values of shear wave velocities from the first time of arrival and zero-crossing methods when *f* < 3.3 kHz, data for specimens (**a**) S1–S10 and (**b**) S11–S12.

**Figure 14 materials-14-00544-f014:**
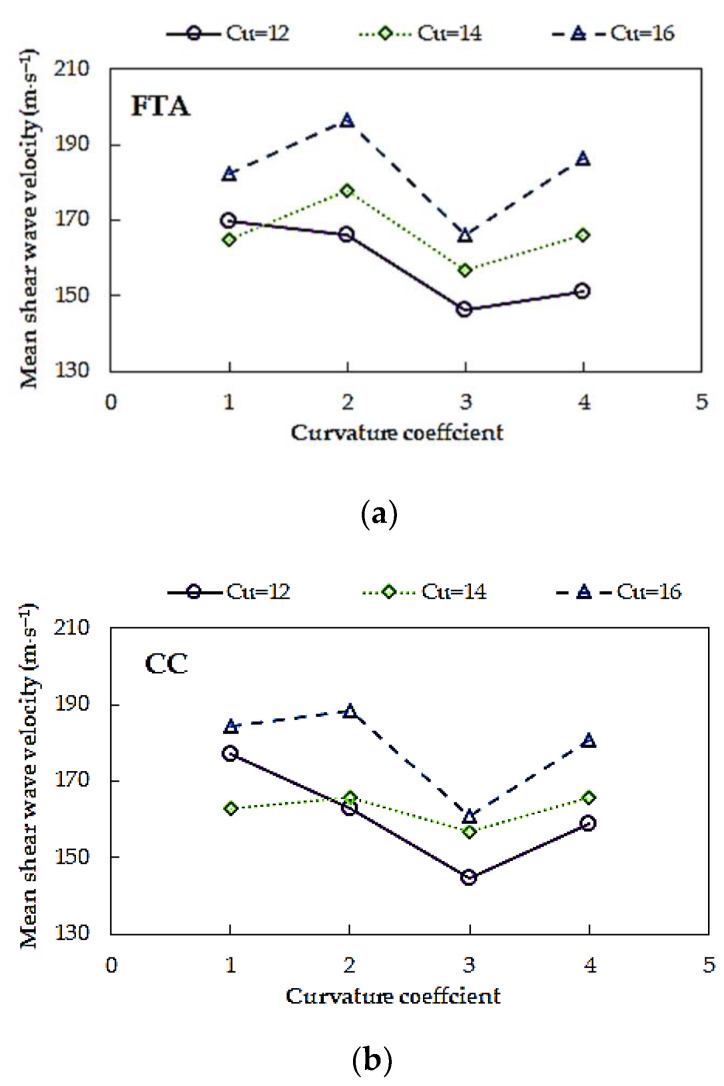
Effect of curvature coefficient (*C_C_*) on mean shear wave velocity for sands of the same coefficient of uniformity (*C_U_*) from: (**a**) first time of arrival method and (**b**) cross-correlation method.

**Figure 15 materials-14-00544-f015:**
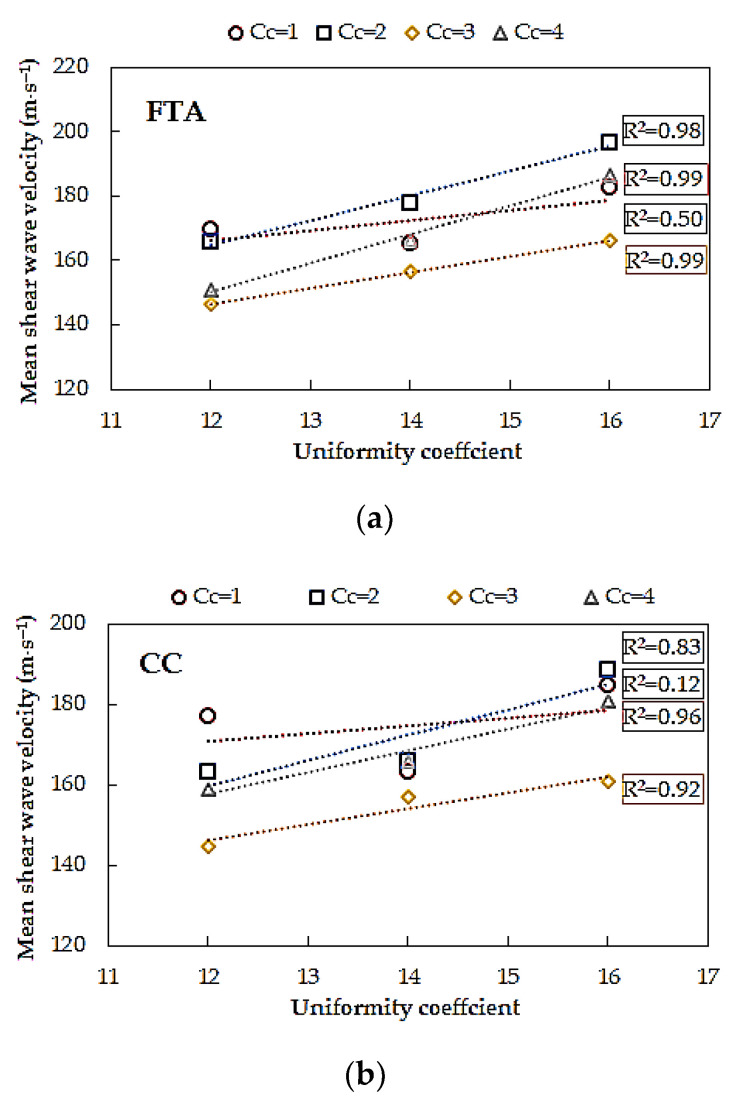
Effect of uniformity coefficient (C_U_) on mean shear wave velocity for sands of the same coefficient of curvature (*C_C_*) from: (**a**) first time of arrival method and (**b**) cross-correlation method.

**Figure 16 materials-14-00544-f016:**
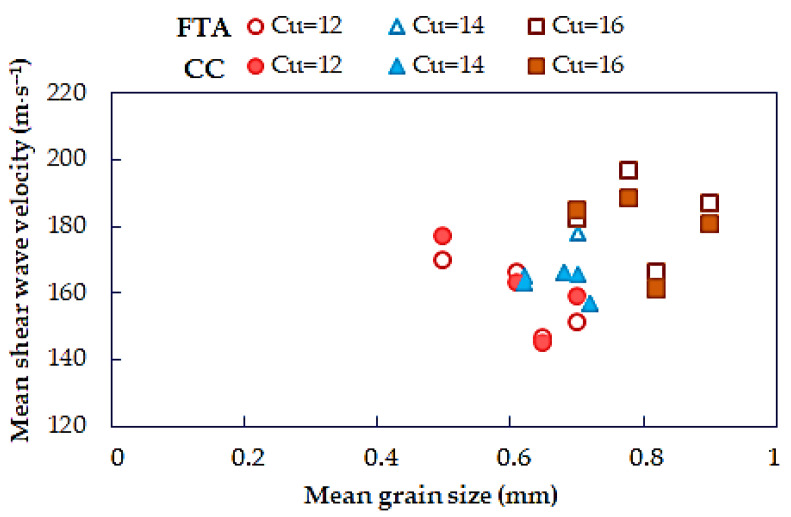
Effect of mean grain size (*d*_50_) on mean shear wave velocity for sands from the first time of arrival method and cross-correlation method.

**Table 1 materials-14-00544-t001:** Basic properties of the soils tested in the experimental investigation.

Sample Name	Gradation							
	*d*_10_ (mm)	*d*_30_ (mm)	*d*_50_ (mm)	*d*_60_ (mm)	e_min_ (-)	e_max_ (-)	m_opt_ (%)	ρ_dmax_ (g/cm^3^)
S1	0.063	0.220	0.50	0.760	0.221	0.568	6.7	2.22
S2	0.063	0.310	0.61	0.760	0.227	0.577	7.3	2.20
S3	0.063	0.379	0.65	0.760	0.233	0.596	8.4	2.17
S4	0.063	0.440	0.70	0.760	0.244	0.596	9.1	2.12
S5	0.063	0.240	0.62	0.885	0.221	0.577	6.8	2.22
S6	0.063	0.340	0.70	0.885	0.244	0.596	7.2	2.20
S7	0.063	0.410	0.72	0.885	0.238	0.596	7.6	2.16
S8	0.063	0.480	0.68	0.885	0.233	0.587	7.5	2.16
S9	0.063	0.250	0.70	1	0.227	0.596	7.3	2.21
S10	0.063	0.355	0.78	1	0.233	0.577	7.1	2.21
S11	0.063	0.435	0.82	1	0.221	0.523	6.9	2.21
S12	0.063	0.500	0.90	1	0.238	0.559	7.0	2.19

**Table 2 materials-14-00544-t002:** Summary of test series.

Test Series	Sample Name	Wave Period (ms); Mean Effective Stress (kPa)	
		(T_S_; *p*′)	(T_P_; *p*′)
I	S1	(0.2, 0.25, 0.4, 0.45; 45)	(0.02, 0.05; 45)
II	S2	(0.2, 0.25, 0.4, 0.45; 45)	
III	S3	(0.2, 0.25, 0.4, 0.45; 45)	(0.05; 45)
IV	S4	(0.2, 0.25, 0.4, 0.45; 45)	(0.02, 0.05; 45)
V	S5	(0.2, 0.25, 0.4, 0.45; 45)	(0.02, 0.05; 45)
VI	S6	(0.2, 0.25, 0.4, 0.45; 45)	(0.02, 0.05; 45)
VII	S7	(0.2, 0.25, 0.4, 0.45; 45)	
VIII	S8	(0.2, 0.25, 0.4, 0.45; 45)	(0.02, 0.05; 45)
IX	S9	(0.2, 0.25, 0.4, 0.45; 45)	(0.02, 0.05; 45)
X	S10	(0.2, 0.25, 0.4, 0.45; 45)	
XI -1	S11	(0.06, 0.08, 0.1, 0.2, 0.3, 0.4, 0.5; 45)	
XI - 2		(0.04, 0.06, 0.08, 0.1, 0.2, 0.3, 0.4, 0.5; 90)	(0.03; 90)
XI - 3		(0.06, 0.08, 0.1, 0.2, 0.3, 0.4, 0.5; 180)	
XII - 1		(0.04, 0.06, 0.08, 0.1, 0.2, 0.3, 0.4, 0.5; 90)	(0.02; 90)
XII - 2		(0.04, 0.06, 0.08, 0.1, 0.2, 0.3, 0.4, 0.5; 180)	(0.02, 0.03; 180)
XIII - 1		(0.04, 0.05, 0.06, 0.08, 0.1, 0.2, 0.25, 0.3, 0.4, 0.45, 0.5; 180)	(0.01, 0.02, 0.03, 0.04; 180)
XIV -1	S12	(0.04, 0.05, 0.06, 0.08, 0.1, 0.2, 0.25, 0.3, 0.4, 0.45, 0.5; 45)	(0.03, 0.04, 0.05, 0.06; 45)
XIV- 2		(0.04, 0.05, 0.06, 0.08, 0.1, 0.2, 0.25, 0.3, 0.4, 0.45, 0.5; 90)	(0.03, 0.04, 0.05, 0.06, 0.08; 90)
XIV - 3		(0.04, 0.05, 0.06, 0.08, 0.1, 0.2, 0.25, 0.3, 0.4, 0.45, 0.5; 180)	(0.03, 0.04, 0.05, 0.06, 0.08; 180)
XV - 1		(0.04, 0.05, 0.06, 0.08, 0.1, 0.2, 0.25, 0.3, 0.4, 0.45, 0.5; 90)	(0.08, 0.1; 90)
XV - 2		(0.04, 0.05, 0.06, 0.08, 0.1, 0.2, 0.25, 0.3, 0.4, 0.45, 0.5; 180)	(0.02, 0.03, 0.04, 0.06, 0.08, 0.1; 180)
XVI - 1		(0.04, 0.05, 0.06, 0.08, 0.1, 0.2, 0.25, 0.3, 0.4, 0.45, 0.5; 180)	(0.04, 0.06, 0.08, 0.1; 180)

## Data Availability

Data available on request due to their size properties. The data presented in this study are available on request from the corresponding author.

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
