# Peer review of "Warsaw Glacial Quartz Sand with Different Grain-Size Characteristics and Its Shear Wave Velocity from Various Interpretation Methods of BET"

_materials, 2021, doi:10.3390/ma14030544_

Round 1

Reviewer 1 Report

Dear Authors,

Please find attached my comments

sincerely

Reviewer 2 Report

The paper presents a rigorous comparative investigation of different techniques to measure the shear wave velocity in sands. In my opinion, it can be published as is. I only recommend to the authors to reread it once more to remove several misprints, e.g.,

Line 186. Commonly, designations are either marked by two commas or not marked at all, depending on the sentence. Here, two commas could be suggested.

Line 190. The same remark. Besides, the space is missed between the formula and the rest of the sentence (should…)

Lines 202, 205, 209. In my opinion, it is better to omit “the” in “different the curvature coefficient »

Line 241. Correct the combination “consists is” in the sentence “The modified triaxial apparatus consists is embedded…”

Line 247. Correct the sentence “…based on the obtained times arrivals of the waves…”

Line 620. “…it should be paid special attention to the methodology…” could be replaced with “special attention should be paid”

Round 2

Reviewer 1 Report

Thanks for your reply on my commenst. vs-Frequency  Figure belongs to Santamarina et al. (2001)'' soils and waves'' book. 

sincerely